# HiBug2: Efficient and Interpretable Error Slice Discovery for Comprehensive Model Debugging

**Muxi Chen**[*] **& Chenchen Zhao**[*] **& Qiang Xu**
Department of Computer Science and Engineering
The Chinese University of Hong Kong
{mxchen21,cczhao,qxu}@cse.cuhk.edu.hk

## Abstract

Despite the significant success of deep learning models in computer vision, they often exhibit systematic failures on specific data subsets, known as error slices. Identifying and mitigating these error slices is crucial to enhancing model robustness and reliability in real-world scenarios. In this paper, we introduce **HiBug2**, an automated framework for error slice discovery and model repair. HiBug2 first generates task-specific visual attributes to highlight instances prone to errors through an interpretable and structured process. It then employs an efficient slice enumeration algorithm to systematically identify error slices, overcoming the combinatorial challenges that arise during slice exploration. Additionally, HiBug2 extends its capabilities by predicting error slices beyond the validation set, addressing a key limitation of prior approaches. Extensive experiments across multiple domains — including image classification, pose estimation, and object detection — show that HiBug2 not only improves the coherence and precision of identified error slices but also significantly enhances the model repair capabilities. Our code is available at `https://github.com/cure-lab/HiBug2`.

## 1 Introduction

Deep learning models have made substantial progress in computer vision tasks. However, they still exhibit systematic failures on critical subsets of data (Buolamwini & Gebru, 2018), known as "error slices". In high-stakes applications like healthcare (Giger, 2018) and autonomous driving (Fujiyoshi et al., 2019; Breitenstein et al., 2021), identifying error slices is crucial to improving model robustness and ensuring safety. At the same time, uncovering error slices in widely-used public models, such as CLIP (Radford et al., 2021) and BLIP (Li et al., 2022a), is also important, as these models are applied across various tasks by a large number of users.

Identifying coherent error slices — subsets of failure samples that share common visual attributes — is challenging due to the lack of detailed visual attribute annotations in most evaluation datasets. Previous works (d'Eon et al., 2022; Yenamandra et al., 2023) typically attempt to identify error slices by clustering failure samples within an embedding space, and relying on human experts to manually annotate coherent slices. Some approaches (Eyuboglu et al., 2022; Jain et al., 2022) incorporate captioning models to assist with slice annotation. We refer to these approaches as "slice-then-tag" methods, where error slice identification is followed by descriptive tag generation. However, these methods often struggle to ensure the coherence of error slices (Gao et al., 2023; Johnson et al., 2023) due to the entangled embedding space (Chen et al., 2024). This makes it difficult for humans to interpret the slices or to conduct efficient model repair.

To address these challenges, particularly with the rise of multi-modal models, a new line of work (Gao et al., 2023; Chen et al., 2024; Liang et al., 2024; Metzen et al., 2023; Gannamaneni et al., 2025) has emerged that prioritizes visual attribute generation before slice discovery. For example, AdaVision (Gao et al., 2023) iteratively uses GPT (OpenAI, 2023) to establish potentially

---

[*]Equal Contribution

critical scenarios and retrieves relevant data for validation, while HiBug (Chen et al., 2024) proposes to exhaustively generate visual attributes for the dataset and cluster the data based on attribute similarity to discover error slices.

While these methods improve slice coherence and offer better interpretability, they still face several limitations. First, the visual attributes used by recent approaches (Eyuboglu et al., 2022; Chen et al., 2024; Liang et al., 2024) primarily focus on object-centric factors such as "object color" and "object type", overlooking contextual elements like background properties, which limits comprehensive error slice discovery. Second, the exploration of attribute combinations can lead to a combinatorial explosion, restricting fine-grained analysis of multi-attribute slices. Furthermore, these methods tend to overlook potential errors beyond the validation set, leaving potential high-risk slices unexplored.

In this paper, we introduce HiBug2, a fully automated, closed-loop debug framework. As depicted in Figure 1, HiBug2 encompasses attribute and tag generation, error slice discovery, and model repair. To generate comprehensive visual attributes, we implement a structured generation process informed by model failure analysis and engineering insights. To mitigate the combinatorial explosion issue, we develop an efficient slice enumeration algorithm based on the unique characteristics of data slices, along with slice selection and image querying techniques to facilitate model repair. Additionally, we also employ feature-based tag substitutions and instruction-based methods to address unseen errors beyond the validation set.

Our experiments across image classification, pose estimation, and object detection tasks, spanning multiple datasets and models, demonstrate the superior performance of HiBug2. Specifically, HiBug2 consistently produces attributes of significantly higher quality than existing methods, while its slice enumeration algorithm achieves an impressive 510x speedup over naive approaches. Furthermore, HiBug2 shows strong generalizability in identifying error slices on widely-used models. For instance, we identified approximately 500 distinct error slices for CLIP in image classification tasks. In addition, the classification models involved in the experiments exhibit performance declines of up to 64.6% on predicted unseen error slices. Finally, experimental results validate that HiBug2 outperforms prior methods in its model repair capabilities.

## 2 RELATED WORKS

### 2.1 ERROR SLICE DISCOVERY

Error slice discovery refers to the process of identifying groups of failure cases in model predictions, akin to failure analysis in engineering disciplines. It helps engineers pinpoint a model's weaknesses and subsequently improve its performance. Due to its practical relevance, interpretability is a key requirement in error slice discovery. The discovered failure groups must be coherent, meaning they share similar and interpretable visual attributes.

**Slice-then-tag methods:** Because of the absence of visual attribute annotations, early methods for error slice discovery (Eyuboglu et al., 2022; Jain et al., 2022; d'Eon et al., 2022; Yenamandra et al., 2023) typically cluster failure samples in an embedding space and then rely on human experts or captioning models to generate descriptions for the identified slices. For instance, Spotlight (d'Eon et al., 2022) locates high-failure areas in the model's embedding space, while FACTS (Yenamandra et al., 2023) amplifies the model's dependencies on latent features and clusters underperforming slices in the CLIP (Radford et al., 2021) space. Domino (Eyuboglu et al., 2022) and Jain et al. (2022) use mixture models and linear classifiers to cluster failures within the CLIP space, and automatically assemble slice descriptions from a predefined natural language corpus. However, studies (Gao et al., 2023; Johnson et al., 2023; Chen et al., 2024) have shown that these methods often struggle to produce coherent slices. Popular embedding spaces (e.g, CLIP) are entangled, making these methods difficult to cluster data based on task-specific attributes (Chen et al., 2024). Additionally, the generated slice descriptions may contain irrelevant or contradictory information (e.g., "a photo of setup by banana", "a photo of skiing at sandal") (Gao et al., 2023), leading to confusion when users attempt to act on the findings (Johnson et al., 2023).

**Tag-then-slice methods:** With the advancement of large multi-modal models, generating attribute annotations has become more feasible. By prioritizing the generation of visual attributes and clustering data accordingly, these methods naturally maintain slice coherence. For example, AdaVi-

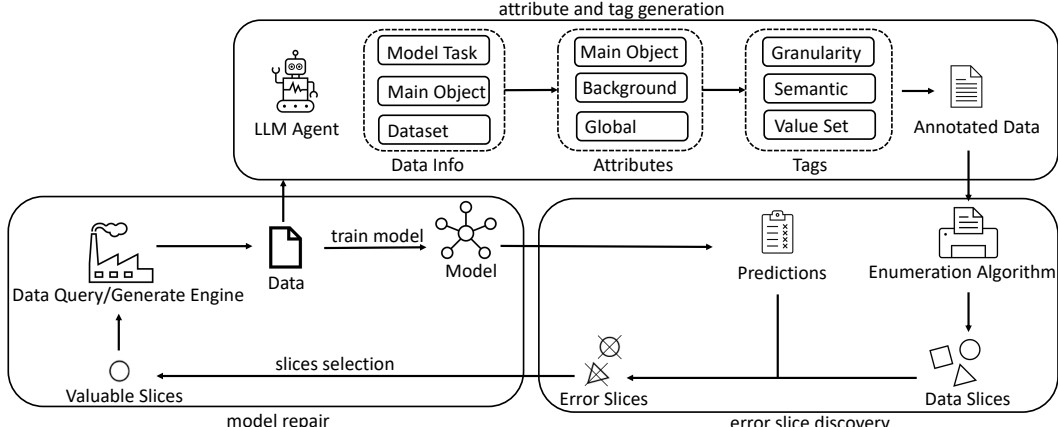

Figure 1: The main workflow of HiBug2 for closed-loop model debugging and repair.

sion (Gao et al., 2023) leverages GPT (OpenAI, 2023) to iteratively generate critical scenarios and retrieve relevant data slices for validation, while Metzen et al. (2023) employ human-defined attributes and tags to generate slices and verify model performance. AIDE (Liang et al., 2024) focuses on object detection, generating descriptions for failure samples and clustering them by object category. HiBug (Chen et al., 2024) proposes an exhaustive approach that generates visual attributes for the dataset and clusters data by attribute similarity. Although these methods offer greater interpretability and coherence, challenges remain with the quality of attributes, and the large number of attributes leads to combinatorial explosions during slice enumeration.

Several methods fall outside these two primary categories. For instance, GradCam (Selvaraju et al., 2017) offers a sample-specific visualization technique to assist humans in discovering error slices, while VLSlice (Slyman et al., 2023) provides an interactive system to retrieve data and validate correlations of interest. SliceLine (Sagadeeva & Boehm, 2021) introduces a monotonic slice scoring function alongside an efficient slice search algorithm.

We adopt a tag-then-slice approach, addressing two key challenges in prior methods: attribute and tag quality, and the efficiency of slice enumeration. Our structured generation process produces task-specific visual attributes that highlight error-prone instance, enhancing the coverage and coherence of identified error slices. Simultaneously, our efficient enumeration algorithm alleviates the combinatorial explosion issue, facilitating the rapid discovery of complex slices that encompass multiple attributes. Additionally, we also address on predicting unseen errors beyond the validation set that are often overlooked in previous works.

## 2.2 VISUAL ATTRIBUTE AND TAG GENERATION

Visual attribute refers to a specific visual characteristic (e.g., "object pose"), while tags are the possible values that describe the attribute (e.g., "standing", "lying down"). Visual attribute and tag generation is fundamental to tag-then-slice approaches, providing the basis for identifying and analyzing error slices. Despite its importance, this area remains underexplored in error slice discovery. Existing methods often rely on human experts (Metzen et al., 2023; Jain et al., 2022; Eyuboglu et al., 2022) or directly query models like GPT with simplistic prompts (Chen et al., 2024), which are inadequate for the complexity of this task.

In other domains, attribute and tag generation has been extensively studied, such as in image tagging and interpretable image classification. However, these methods are not applicable to error slice discovery. Image tagging models (Zhang et al., 2024; Chen et al., 2023) are designed to identify image content and generate image-specific tags; however, the significant variation in these tags across different images limits the ability to form coherent data slices. Similarly, approaches in interpretable image classification (Yang et al., 2023; Yan et al., 2023) generate attributes to differentiate between classes, whereas error slice discovery requires attributes that capture failure patterns and cause confusion between image classes, presenting a unique challenge.

## 3 METHOD: AUTOMATIC ATTRIBUTE AND TAG GENERATION

We present the workflow of HiBug2 in Figure 1. Attribute and tag generation serves as the foundation of HiBug2, as it is closely linked to the coherence and coverage of error slices. Our attribute and tag generation consists of the following steps: attribute generation, tag determination, and dataset-wide tag assignment. It addresses several key challenges identified in existing approaches.

### 3.1 KEY CHALLENGES IN ATTRIBUTE AND TAG GENERATION

Current methods (Chen et al., 2024; Liang et al., 2024; Eyuboglu et al., 2022) for generating attributes and tags for image datasets exhibit several critical shortcomings:

1. **Narrow Attribute Focus**: Existing methods primarily concentrate on attributes related to the main objects of interest in the images, often overlooking crucial contextual factors such as background properties and global image characteristics. Furthermore, these attributes tend to be general rather than specifically tailored to error-related and task-specific needs.
2. **Inconsistent and Biased Tagging**: Tags are generated directly from the data without external references. Due to the biases of the data, the generated tags often have inconsistencies in granularity and semantics for the same attribute.

### 3.2 STRUCTURED AND COMPREHENSIVE GENERATION

To address these challenges, HiBug2 leverages the strengths of multi-modal models (in this paper, we use GPT (OpenAI, 2023)) for attribute and tag generation, combined with a structured process to ensure the accuracy, consistency, and coverage of the results.

#### 3.2.1 ATTRIBUTE GENERATION

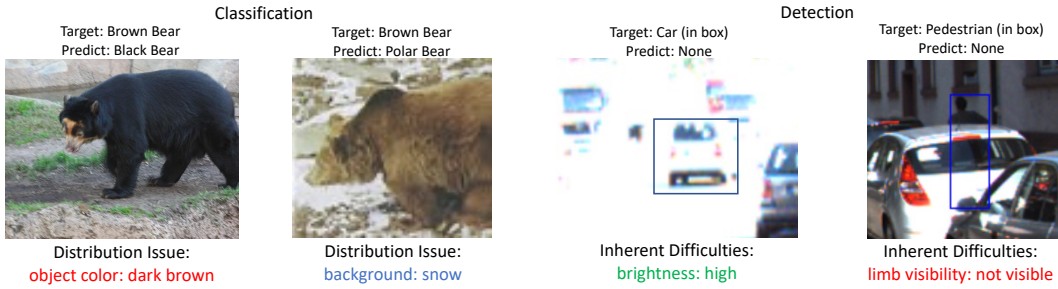

Figure 2: Common errors of deep learning models can be grouped into data distribution issues and inherent task difficulties.

To effectively capture the properties related to error slices, the generated attribute set is required to cover a diverse range of image characteristics that influence model performance.

Through interviews with engineers and a detailed analysis of common errors in classification and object detection models, as illustrated in Figure 2, we identify two primary types of model errors: errors caused by data distribution issues (e.g., rare cases, distribution shifts, spurious correlations, etc), and errors resulting from inherent task difficulties (e.g., occlusions, small object sizes, low image resolutions, etc). In response to these error sources, we categorize attributes into three key types according to the subjects they refer to: *main object*, *background*, and *global*. This structured categorization ensures that we capture not only the features of the primary object but also essential contextual and global information of the images.

*Main object attributes* capture properties that are directly related to the objects of interest, such as their shapes and colors. These attributes are crucial in tasks such as classification, where the model primarily focuses on distinguishing features of the main objects. *Background attributes* refer to elements surrounding the main objects, such as the environment and distracting objects in the scene, which are particularly important in tasks such as object detection. *Global attributes* describe overall

image characteristics, such as resolutions, noise levels, and lighting conditions, which can introduce image-wide artifacts and impact model performance across a variety of tasks.

Existing methods often rely on directly querying LLMs with generic questions such as "Give me some attributes related to humans". Given the potentially infinite number of attributes, this approach is inefficient and unlikely to capture the subtle distinctions necessary for identifying error slices. To tackle this challenge, we develop two targeted attribute generation algorithms respectively for the two error types presented above. To address errors caused by data distribution issues, on top of direct query, we further employ a comparative approach. Specifically, we present multi-modal models with a sufficient number of image pairs from the dataset and ask them to generate contrasting attributes that highlight key differences within the pairs. This ensures that the generated attributes are not only comprehensive but also contextually grounded in the dataset, helping to uncover subtle biases and anomalies. To capture attributes associated with inherent task difficulties, we design task-specific queries aimed at identifying error-prone features. For example, in classification tasks, we instruct the model to generate attributes that could blur the boundaries between two similar categories if assigned with specific tags. In object detection and pose estimation tasks, we focus on identifying features that may lead to localization errors or issues with occlusions.

With this structured and targeted approach for attribute generation, we ensure that the generated attributes are both relevant and capable of addressing the error-inducing factors present in the dataset.

### 3.2.2 Tag determination and assignment

Once the attributes are determined, the next step is to generate consistent and meaningful tags for each attribute. A common issue in previous works is the inconsistency in tag generation in terms of both granularity and alignment with the semantic definitions of the attributes. To mitigate this issue, our method employs a multi-stage refinement process that operates as follows.

HiBug2 begins by generating an initial list of potential and unbiased tags for each attribute, establishing the semantic scope and granularity. To ensure that the generated tags are concrete and unambiguous, we employ clear conventions: for binary attributes, such as "whether an object is occluded", we adopt a straightforward "yes/no" tag format, while for open-ended attributes, Hi-Bug2 generates a descriptive set of tags. Next, HiBug2 collects new tags by reviewing a subset of validation data, incorporating additional tags as needed to ensure comprehensive coverage of variations. Once the attributes and tags are finalized, HiBug2 assigns these tags to all images within the dataset, ensuring that each image receives a tag corresponding to each attribute. This multi-stage process promotes consistency in tags throughout the dataset, thereby enhancing the robustness of subsequent analyses. We discuss HiBug2's scalability and potential issues regrading attribute and tag generation in Appendix A.8.

## 4 Method: exploring data slices

### 4.1 Efficient slice discovery and repair

With the generated attribute and tag sets, we define "data slices" as data subsets that share a tag or a combination of tags from several attributes. Slice enumeration aims to comprehensively enumerate data slices, forming the foundation for error analysis and subsequent model repair.

### 4.1.1 Basic notations

Let $A = \{a_1, a_2, \ldots, a_n\}$ be a set of attributes, where each attribute $a_i$ has a corresponding set of possible tags $T_i = \{t_{i1}, t_{i2}, \ldots, t_{im_i}\}$. Slice $S$ is defined as a combination of tags from these attributes, where each attribute contributes one tag to the slice:

$$S = \{(a_1, t_{1j_1}), (a_2, t_{2j_2}), \ldots, (a_k, t_{kj_k})\}, \text{ with } t_{ij} \in T_i, k \leq n \tag{1}$$

### 4.1.2 Key challenges in slice enumeration

Let the number of tags within a set $T_i$ be denoted as $|T_i|$. The primary challenge in slice enumeration arises from the combinatorial explosion of potential slices, which is upper-bounded by the term:

$\sum_{i=1}^{n} \binom{n}{i} (\max |T_i|)^i$. When considering combinations of $k$ attributes within a validation set of $N$ data points, the brute-force enumeration process has a time complexity of: $\sum_{i=1}^{k} \binom{n}{i} (\max |T_i|)^i \times N$. This complexity increases rapidly with the number of attributes in a slice $k$, the total number of attributes $n$, and the size of the validation set $N$. As a result, exploring error slices involving multiple attribute combinations becomes computationally infeasible without efficient algorithms.

### 4.1.3 PROPERTIES OF DATA SLICES

Data slices are characterized by two key properties: the *average model performance* and the *data count*. The average model performance (e.g., average accuracy in image classification, average object keypoint similarity (OKS) in pose estimation, and average intersection-over-union (IoU) in object detection) is essential for identifying error slices, as it reflects the model's performance on a specific subset of the data. The data count indicates the prevalence of the slice and is related to the reliability and generalization of the error pattern. In general, a model's performance on slices with larger data counts is more likely to generalize to unseen data belonging to those slices.

**Monotonicity**: During enumeration, we define monotonicity by comparing a slice $S$ with its parent slice $S_p$, where $S_p \subset S$. The average model performance is *non-monotonic*, as it may increase or decrease when moving from a parent slice to a child slice. In contrast, the data count is *monotonic*, as it always decreases or remains constant from parent to child slices.

### 4.1.4 ALGORITHM DESIGN

**Breadth-First Tree-structured Enumeration.** To address the above challenges, we propose a breadth-first tree-structured enumeration process. We establish a tree based on the generated attributes and tags, in which slices stored in child nodes are formed by adding one attribute-tag pair to slices stored in their parent nodes. The parent-child relationships of slices can then be represented by the parent-child relationships of nodes, and the numbers of attributes in the slices grow with the tree's depth. Note that a child node may correspond to multiple parent nodes in the tree. Crucially, since the data count monotonically decreases with deeper layers, data enumeration of a slice can be upper-bounded by its parent slices, significantly reducing the search space. We employ breadth-first search (Bellman, 1958) (BFS) to ensure that parent slices are enumerated before child slices.

**Pruning and Intersection.** Uninformative slices, particularly those with low data counts, offer limited insight into model errors. Since the data count decreases monotonically with deeper layers, we can safely prune subtrees with the data counts of the root nodes (i.e., slices) fewer than $M$ (in our experiments, $M = 10$).

Similarly, due to the monotonicity of slice data counts, a necessary condition for any informative slice is that all of its parent slices must be retained. Therefore, rather than generating all possible slices for each new layer, we intersect the slices that survive pruning from the previous layer to form new candidates. We define two slices in the same layer as a matched pair if they share $k - 1$ attributes, where $k$ is the number of attributes in the slices of the current layer. By intersecting these matched slice pairs, we only maintain new slices that are likely to yield informative insights. Additionally, we use hash tables for fast-matched pair search and matrix multiplication to speed up data counting and accuracy calculations. The pseudo code is in Appendix A.7.

**Post-processing.** Unnecessary tags that are not related to errors can be confusing when presented in error slices. Therefore, after enumeration, we conduct post-processing to remove slices that have higher average model performance than their parent slices. Notably, slice enumeration is dataset-specific and only needs to be executed once for all models on the same dataset, whereas the post-processing step must be performed individually for each model.

**Integration with Model Repair.** We incorporate image querying techniques for model repair. After post-processing, HiBug2 first ranks the slices based on their average model performance. Given a data pool, HiBug2 then assigns tags to the data and prioritizes those corresponding to error slices with the lowest average performance. This ensures that the most critical cases are addressed first, effectively targeting the model's weakest points for repair.

## 4.2 PREDICTING ERROR SLICES BEYOND THE VALIDATION SET

The validation set may not capture all potential error types and their corresponding data slices, leaving some high-risk slices unfixed in data repair. To mitigate this, we propose two strategies for predicting potential error slices. The predicted slices will serve as complements of discovered error slices when the validation set is limited in size.

**Tag Substitution.** We compute the text embeddings for all tags using CLIP (Radford et al., 2021). For each identified error slice, one of its tags is substituted with another tag from the same attribute that has the closest feature distance. This method is akin to data augmentation, allowing exploration of nearby regions in the feature space from the existing identified regions.

**Instruction-Based Method.** We utilize few-shot learning with GPT to predict potential error slices based on the provided attributes and tags, following task-specific instructions. These instructions are similar to those used for generating error-related attributes. For instance, in image classification, we instruct the model to generate slices that blur boundaries between closely related categories. In object detection and pose estimation, we focus on generating slices prone to localization errors or occlusions. Unlike Tag Substitution which is built upon identified error slices, the instructions-based method leverages GPT's extensive knowledge base without any prior information about the model.

## 5 EXPERIMENTS

### 5.1 COMPARISONS OF ATTRIBUTE AND TAG GENERATION

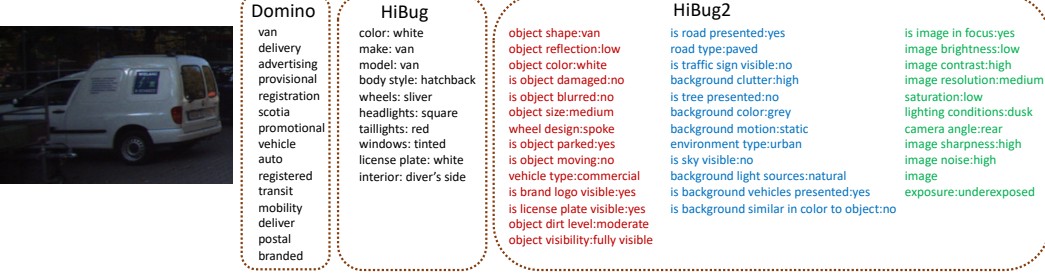

Figure 3: Attributes and tags generated by the error slice discovery methods.

In this section, we present detailed comparisons of the attributes and tags generated by different error slice discovery techniques. All results are generated without human intervention. As illustrated in Figure 3, when applied to an image of a *van* in an object detection scenario, Domino produces overly generic descriptions such as "van" or "delivery", offering little insight into the visual characteristics or potential error patterns. Furthermore, the unstructured nature of these tags limits their usefulness for systematic error analysis. HiBug improves upon this by generating more structured tags, such as "color: white" and "make: van", but it focuses primarily on surface-level characteristics, neglecting important background and global attributes.

In contrast, our method generates attributes and tags that are both task-specific and highly relevant to error slice discovery. We capture not only essential object properties such as "object shape: van" and "object color: white" but also more nuanced details, such as "is object damaged: no" and "object visibility: fully visible". Additionally, our method considers environmental factors and image quality indicators, such as "background clutter: high" and "image sharpness: high", which are crucial for diagnosing model failures in real-world scenarios.

Moreover, in the context of error slice discovery, precise dataset annotations are critical. We observe that attributes generated by HiBug are often ambiguous, leading to inconsistent tag semantics across the dataset. For instance, tags under "wheel" refer to both wheel shape and color, creating confusion. In contrast, our method generates clear, structured attributes and tags, ensuring consistency in both semantics and granularity. Overall, the attributes and tags generated by our method are more effective for model debugging and refinement compared to existing approaches. We provide a full list of the generated attributes and tags by our method and the baselines in Appendix A.2

## 5.2 EFFECTIVENESS OF SLICE ENUMERATION

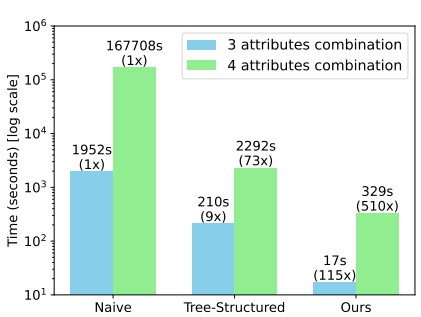

Figure 4: Comparison of the slice enumeration methods.

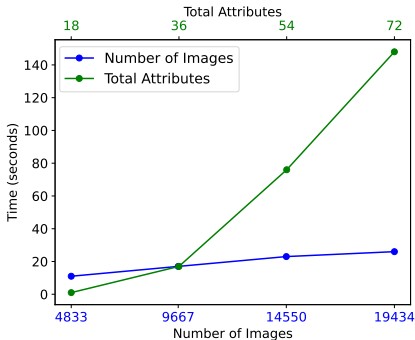

Figure 5: Ablation study across varying numbers of images and tag sets.

We conduct experiments to evaluate the performance of our slice enumeration algorithm in comparison with both a naive enumeration approach and a baseline version of the breadth-first tree-structured algorithm. The naive approach is a brute-force method that lists all possible data slices and searches for matching data for each slice, as described in Section 4.1.3. The breadth-first tree-structured baseline refers to a simplified version of our slice enumeration algorithm, without pruning and intersection. We present the algorithms of the three methods in the Appendix A.7.

As shown in Figure 4, our method achieves approximately 115x and 510x speedups over naive enumeration, and 12x and 7x speedups compared to the baseline tree-structured algorithm, for the enumeration of slices with 3 and 4 attributes respectively. These significant improvements in efficiency allow for rapid slice enumeration across multiple attributes, substantially reducing the computational time required for model analysis in real-world scenarios. Additionally, we conduct ablation studies focusing on the enumeration of slices with 3 attributes to assess the effect of data volume and total number of attributes in runtime. The results in Figure 5 demonstrate that runtime increases linearly with the data volume, while remaining feasible even in cases where tasks involve up to 72 attributes.

## 5.3 IDENTIFIED ERROR SLICES

We conduct experiments across three tasks — image classification, pose estimation, and object detection — to evaluate our method's ability to identify error slices. For image classification, we select five bear species from ImageNet (Deng et al., 2009) and debug three models: ResNet18 (He et al., 2016), CLIP (Radford et al., 2021), and BLIP (Li et al., 2022a). For pose estimation, we use an industrial private dataset and debug the tiny, small, medium, and large variants of RTM-Pose (Jiang et al., 2023). For object detection, we use the "Car" and the "Pedestrian" instances of the KITTI (Geiger et al., 2012) dataset and debug four models: YOLOv8 (Varghese & Sambath, 2024), CO-DINO (Zong et al., 2023), ViTDet-L (Li et al., 2022b), and RTMDet-X (Lyu et al., 2022). We apply HiBug2 to automatically generate attributes and tags and perform slice enumeration, followed by error slice analysis based on the results. In the main paper, we consider slices with three attributes. Implementation details of all models and datasets are shown in Appendix A.3.

Error slices are defined as slices with average performance value (i.e. accuracy, OKS and IoU for the three tasks) lower than the overall model performance by a constant $C$ (in our experiments, $C = 0.2$). For image classification, we identify 1086, 499 and 384 error slices for ResNet18, CLIP and BLIP respectively. Figure 6 showcases error slices of these models, revealing model-specific weaknesses. For example, the first slice for "teddy bears" suggests that ResNet18 struggles with distinguishing "white" and "not holding item" teddy bears with "polar bears". Similarly, the third and fourth slices shows the potential dependency of CLIP and BLIP on colors in classifying bears. For pose estimation, we identify 11357, 5159, 2259, and 2053 error slices for the tiny, small, medium, and large RTMPose models respectively, with common errors arising when people are lying down, wearing black clothes, or crossing their legs. For object detection, we identify 4808, 2918, 2258, and 2014 error slices for YOLOv8, RTMDet-X, ViTDet-L, and CO-DINO respectively. Figure 7 presents low- and high-IoU slices for cars and pedestrians.

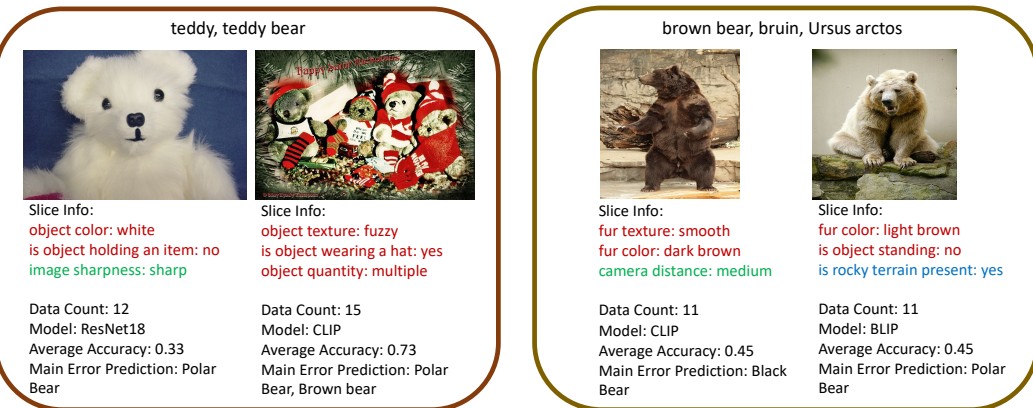

Figure 6: Identified slices of the image classification task by HiBug2.

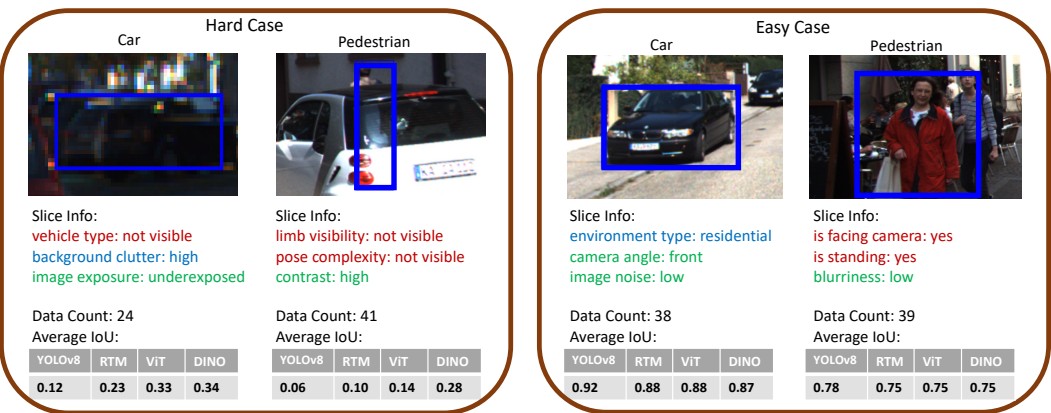

Figure 7: Identified slices of the object detection task by HiBug2.

Interestingly, we observe a clear trend in the object detection task: across both hard and easy cases, all models demonstrate consistent performance patterns. To validate this observation, we compute the overlap of the top 10% slices (i.e., slices with the lowest 10% average IoU) among the four object detection models. The average overlap is 86%, indicating that these models share similar failure patterns, likely due to inherent task difficulties in object detection. Similarly, the average overlap among the pose models is 73%. In contrast, the top 10% error slices for the three classification models overlap by only 31%, suggesting that classification failures are more influenced by data distribution issues. These experiments highlight the effectiveness of our method in identifying and analyzing error slices across diverse tasks, offering valuable insights for improving model performance. More visualizations of the identified error slices are presented in Appendix A.4.

## 5.4 PREDICTIONS OF UNSEEN SLICES

In this experiment, we use ResNet18, RTMPose-Tiny, and YOLOv8 respectively for the image classification, pose estimation, and object detection tasks. We begin by obtaining the predicted error slices using both the tag substitution and instruction-based methods across all the three tasks. These methods generate 100, 20, and 40 slices respectively for image classification, pose estimation, and object detection (with 20 slices per class; pose estimation having only the *person* class). Subsequently, we compute the model's performance on the predicted slices.

The results presented in Table 1 demonstrate that the model's average performance on these predicted error slices is significantly lower than its overall performance. This highlights the effectiveness of our approach in predicting extra errors slices, which is particularly valuable when the validation set available for model debugging is limited in size. Meanwhile, the results further demonstrate the high quality of our generated attributes and tags, particularly in describing potential model errors.

Table 1: Model performance degradation on the predicted error slices by the two proposed methods

| Method | Image Classification | Pose Estimation | Object Detection |
|---|---|---|---|
| Tag Substitution | -7.6% | -34.5% | -64.6% |
| Instruction-Based | -18.4% | -27.2% | -18.0% |

Table 2: Comparisons of model improvements with data slices determined by HiBug2, HiBug, and random selection. The values are averaged over five runs.

| Method | Image Classification | Pose Estimation | Object Detection |
|---|---|---|---|
| HiBug2 (ours) | **0.839 (+7.6%)** | **0.839 (+2.1%)** | **0.683 (+4.9%)** |
| HiBug | 0.832 (+6.3%) | 0.831 (+1.1%) | 0.673 (+3.4%) |
| Random | 0.817 (+4.7%) | 0.829 (+0.9%) | 0.655 (+0.6%) |
| Original | 0.780 | 0.822 | 0.651 |
| Number of extra data | 0.5K (10%) | 1.24K (5%) | 0.5K (fine-tuning) |

## 5.5 MODEL REPAIR

In this section, we evaluate the model repair capabilities across the three previously discussed tasks, focusing on querying new data based on the identified error slices for model improvement. For each task, we construct a query set from which additional data is selected, and the model is evaluated on a hold-out test set, both distinct from the validation set used for identifying error slices. We compare HiBug2 with HiBug (Chen et al., 2024), which is also an automated method, as well as random data selection. For a fair comparison, we implement the same data selection strategy for both HiBug2 and HiBug, prioritizing data that corresponds to slices with the lowest average model performance. The model we repair for the three tasks are respectively ResNet18, RTMPose-Tiny, and RTMDet-X. Implementation details can be found in Appendix A.5.

We summarize the model's improvements in terms of accuracy, keypoint average precision (AP), and object mean average precision (mAP) for the three tasks in Table 2. HiBug2 consistently outperforms other methods; notably, it enhances model performance when random data selection yields only marginal improvements. This underscores its effectiveness in model repair.

## 6 DISCUSSION

Our experiments demonstrate that HiBug2 significantly advances error slice discovery. It improves both coherence and coverage of identified data slices, which leads to a more interpretable and insightful error analysis process. Meanwhile, the efficient slice enumeration algorithm allows for rapid discovery of slices across multiple attributes, enabling a more granular analysis of model errors.

However, there are some limitations associated with HiBug2. Attribute and tag generation leverages GPT that may occasionally produce errors. A primary concern might be the impact of incorrect tag assignments. Though such errors may slightly affect the coherence of error slices, they are unlikely to influence overall identification, as a few misclassified data points do not alter the average performance of a slice. We further discuss several potential issues and the scalability of HiBug2 in Appendix A.8. Moreover, we observe that existing slice discovery methods follow diverse workflows, complicating direct and equitable comparisons. Future work can focus on developing a standardized and widely applicable benchmark to facilitate fair evaluations and drive progress in this area.

## 7 CONCLUSION

In this paper, we introduce HiBug2, a comprehensive framework for efficient and interpretable error slice discovery aimed at enhancing model debugging and repair. By leveraging task-specific attribute generation, efficient slice enumeration, and prediction of unseen error slices, HiBug2 significantly improves coherence and coverage of identified slices across diverse tasks, including image classification, pose estimation, and object detection. Our extensive experiments demonstrate that HiBug2 not only outperforms existing methods in terms of error slice discovery and model repair but also provides deeper insights into model failures. We believe HiBug2's capabilities can drive broader adoption of slice-based debugging techniques in both academic and industrial settings.

## 8 ACKNOWLEDGMENTS

This work was supported in part by the CUHK Strategic Seed Funding for Collaborative Research Scheme under Grant No. 3136023.

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

# A   APPENDIX

## A.1   SCALABILITY AND ROBUSTNESS OF HIBUG2

We address potential concerns regarding the scalability and robustness of HiBug2, particularly focusing on the attribute and tag generation process. We hope that this discussion provides additional clarity and insight into the scalability and robustness of HiBug2. We will also add alternative versions of HiBug2 with reliance on other multi-modal models, such as LLaVA (Liu et al., 2024) and QWen-VL (Bai et al., 2023), in the future.

### A.1.1   ENSURING CORRECT ATTRIBUTE AND TAG GENERATION:

Although our method leverages GPT (OpenAI, 2023) for generating attributes and tags, we implement several mechanisms to ensure correctness. First, we have designed an extensive set of rules to validate the generated outputs and handling exception scenarios. For example, during dataset-wide tagging, we verify that the names of the generated attributes and tags align with the predefined categories and that each tag belongs to the appropriate tag set. Additionally, we introduce the tag "not visible" for attributes related to elements that may not be present in every image, such as background features, thereby enhancing flexibility and accuracy in handling various scenarios.

Second, we employ self-correction loops to refine the generation process. For instance, during attribute generation, we instruct GPT to validate the generated attributes, ensuring there is no conceptual overlap or inappropriate attributes for the objects of interest at hand.

Finally, our prompting strategy incorporates a few-shot approach, featuring carefully curated examples (distinct from the cases in our experiments). We have observed that this few-shot strategy significantly improves performance. During our experiments, we encountered minimal issues regarding the correctness of the outputs.

### A.1.2   INFLUENCES OF TAGGING BIASES:

The tagging process is analogous to a multi-label classification task. Although GPT-4 generally performs well, occasional tagging errors may occur. These errors might slightly affect the coherence of certain error slices, but they are unlikely to impact the overall identification process. Since slice identification is rooted in statistical analysis, the presence of a few misclassified data points typically does not alter the average performance of a slice.

### A.1.3   HANDLING UNNECESSARY ATTRIBUTES:

It is difficult to assess the necessity of specific attributes prior to the error slice discovery phase. HiBug2 generates a substantially larger set of attributes than existing methods, some of which may not directly align with human recognition. However, only after analysis can the utility of these attributes be determined. For instance, in our classification analysis using BLIP, an attribute such as "is man-made object presented" may initially appear irrelevant. Yet, error slice discovery reveals that BLIP tends to misclassify black bears as sloth bears when man-made objects, such as the iron fence of the zoo, is present in the image. Furthermore, we have developed a post-processing algorithm that effectively removes unnecessary attributes during the error slice discovery stage, ensuring that only the relevant attributes are retained for slice analysis.

### A.1.4   GENERALIZATION TO DIFFERENT TASKS AND DATASETS:

Extending HiBug2 to other tasks primarily involves adjusting the attribute and tag generation process, particularly the task-specific prompts used for generating relevant attributes. We provide prompt templates for easy task extension, allowing users to extend our method by simply modifying task descriptions. For different datasets within the same task, the primary variation lies in the change of the main object. Through additional experiments on the full ImageNet and COCO detection datasets (not included in the paper), we observe that HiBug2 generalizes well across a wide range of objects, demonstrating its scalability and flexibility for various tasks and datasets.

A.1.5 TIME COST OF HIBUG2.

The most time-consuming part of HiBug2 is the process of tag assignment for all the images in the dataset, which typically accounts for over 95% of the total time, with larger datasets increasing this proportion. It operates with $O(N)$, where $N$ is the number of images in a dataset, because GPT-4V generate all attributes for one image in one step. In our experiments, a single query takes around 6 seconds. We implement parallel processing to accelerate these processes, since the tag assignments for different data are independent. For a dataset of 10,000 images, if we use 50 threads to accelerate the labeling process, it typically takes around 30 minutes to run HiBug2. It is also worth noting that attribute and tag generation and slice enumeration only need to be performed once for a dataset. When iteratively improving the performance of a model, only the first round requires labeling and slice enumeration. Subsequent iterations only require the post-processing steps described in Section 4.1.4, which takes only a few seconds.

A.1.6 THE DIFFICULTIES IN EVALUATION.

Unlike other fields, we cannot find a standardized evaluation process for error slice discovery methods. Studies often design custom-built datasets and conduct limited comparisons. We identify two key challenges:

1. Differences in workflow. Even when methods aim for similar goals, as mentioned in section2.1, some methods rely heavily on human analysis with LLM-assisted slice discovery, while others cluster data before human labeling. Approaches like ours and HiBug, however, label the data first and then discover slices based on attribute clustering. These workflow differences make it challenging to establish suitable baselines for evaluating each component of the method.

2. Error slices have no ground truths. Error slices require shared human-understandable attributes, but even for the same group of data, different individuals may define and label attributes in vastly different ways(Johnson et al., 2023).

A.1.7 ARE ERROR SLICES DISTINCT BUGS?

Table 3: Model performance on different kinds of data before and after fixing one error slice.

| Data type | Match | Overlap | Non-overlap |
|-----------|-------|---------|-------------|
| Before    | 0.500 | 0.840   | 0.829       |
| After     | 0.875 | 0.862   | 0.791       |

In this paper, error slices are defined based on the combination of tags, and therefore, slices may overlap in some tags. While error slices are distinct in terms of their tag combinations, they do not necessarily correspond to distinct bugs. Fixing one error slice may improve the model's performance on other related error slices as well.

To test this, we designed a simple experiment to check the impact of fixing one error slice on others. For the ResNet model used in the classification task described in the main paper, we chose an error slice corresponding to class teddy bear, defined as (object color: white, object pose: sitting). We then queried the data matching this error slice (20 images) from a hold-out dataset. We fine-tuned the model on this data for 20 epochs and evaluated its accuracy on the validation set. We collected the model's accuracy on three types of data: (1) Data matching the selected error slice. (2) Data belonging to error slices that overlap with the selected slice (e.g., (object color: white, xxx), (xxx, object pose: sitting)). (3) Data belonging to error slices that does not overlap with the selected slice.

The results in Table 3 reveal a significant improvement in the model's performance on the fixed error slice. Interestingly, performance on overlapping error slices also improved, suggesting a potential relationship between these slices and shared model weaknesses. However, performance on non-overlapping data decreased slightly, likely due to overfitting to the specific distribution of the fine-tuning data. This experiment suggests that while error slices are defined by distinct tag combinations, they are not distinct bugs.

## A.2 GENERATED ATTRIBUTES AND TAGS BY HIBUG2 AND THE BASELINE METHODS

Table 4: The full list of attributes and tags generated by **the baseline HiBug** for the pose estimation task. For tag sets containing more than five tags, the remaining tags are omitted with ellipses

| Attributes | Tags |
|---|---|
| hair color | gray, white, red, black, ... |
| eye color | red, blue, black, brown, ... |
| clothing | coat, jacket, underwear, ... |
| facial expression | smiling, excitement, sad, ... |
| height | 1 foot, 5'8, 3 feet, ... |
| posture | laying down, squatting, running, ... |
| age | young, teens, 12, ... |
| accessories | no, earrings, yes, ... |
| skin tone | light and dark, gray, Asian, ... |
| surroundings | field, kitchen, garage, ... |

We present a full list of attributes and tags generated by HiBug2 and HiBug for pose estimation in Table 4 and Table 5. The attributes generated by HiBug primarily focus on the main object, often neglecting task-specific requirements and potential errors. Furthermore, the tags frequently lack semantic consistency; for example, the tags associated with the attribute "age" include overlapping terms such as "young", "teen", and "12". Similarly, the tags for "skin tone" combine both race and color categories, resulting in semantic ambiguity. This inconsistency can adversely affect the data slicing process, leading to images with similar semantics being assigned to different slices (e.g., slice "teen" versus slice "young"). In contrast, the attributes and tags generated by HiBug2 are specifically tailored to tasks and errors, ensuring semantic consistency. This makes HiBug2 more effective for model debugging and refinement.

## A.3 DETAILED EXPERIMENTAL SETUP OF SLICE IDENTIFICATION

For image classification, we combine the original training and validation sets of ImageNet. Our error slice identification focuses on the last 850 images per class, while the first 500 images are used to train ResNet-18. CLIP (ViT-H-14) and BLIP are public models that have not been explicitly trained on ImageNet. For these models, we perform zero-shot classification by comparing image features with text features representing class names.

For pose estimation, our pose dataset contains 47,057 images (24,832 from COCO, with the remainder from a private source), primarily used for rehabilitation training in hospitals to recognize patient movements and assess whether exercises meet required standards. In our experiments, models are pre-trained on the COCO portion. For error slice discovery, we use 7,057 images from the private portion as the validation set. For model repair, we select 1,241 images from the private portion.

For object detection, YOLOv8 is trained on the first 5000 images of the KITTI training set. The other models are pretrained on COCO and are set to focus only on car and pedestrian predictions. The remaining 2481 images from the dataset are used for error slice identification.

## A.4 MORE CASES OF THE IDENTIFIED ERROR SLICES

We present additional visualizations of the identified error slices, along with some noteworthy observations in Figure 8, Figure 10 and Figure 12. For instance, in Figure 8, we observe that BLIP (Li et al., 2022a) struggles to correctly classify most black bears whose fur color closely resembles brown. This suggests that the model's predictions may rely heavily on color rather than on biological features such as ear shape, body size, or the curvature of the bear's back. This poses significant challenges for researchers developing animal classification applications based on BLIP.

We also provide the error slices identified by HiBug for comparison in Figure 9 and Figure 11. Notably, the vanilla HiBug focuses only on error slices defined by a single attribute. To enable slices with multiple attributes, we applied our slice enumeration algorithm to HiBug. We observe that the quality of attributes and tags significantly impacts the identified error slices. HiBug's attributes focus solely on the main object, and tags for certain attributes, such as 'eyes' in Figure 9, show inconsistencies (e.g., 'round' for shape and 'black' for color). These limitations can negatively affect the user experience in practical applications.

Table 5: The full list of attributes and tags generated by **HiBug2** for the pose estimation task. For tag sets containing more than five tags, the remaining tags are omitted with ellipses

| Main Object | |
|---|---|
| **Attributes** | **Tags** |
| is arm crossing | yes, no |
| pose complexity | simple, medium, complex, not visible |
| clothes color | red, blue, green, yellow, ... |
| is standing on one leg | yes, no |
| is carrying something | yes, no |
| is on all fours | yes, no |
| pose | sitting, jumping, lying down, ... |
| head orientation | front, back, sideways, ... |
| size | large, medium, small |
| object orientation | upright, sideways, inverted, ... |
| is sitting | yes, no |
| is using props | yes, no |
| leg position | together, apart, crossed, ... |
| limb visibility | both arms visible, one arm visible, ... |
| is crouching | yes, no |
| is partially occluded | yes, no |
| clothes type | casual, formal, sportswear ... |
| facial expression | smiling, frowning, neutral, ... |
| is holding hands behind back | yes, no |
| is leg crossing | yes, no |
| clothes fit | tight, loose, fitted, not visible |
| **Background** | |
| **Attributes** | **Tags** |
| is sky presented | yes, no |
| clutter | high, medium, low |
| is natural habitat presented | yes, no |
| background style | urban, rural, natural, artificial, indoors |
| indoor lighting | bright, dim, natural, ... |
| is dynamic | yes, no |
| is containing other people | yes, no |
| is background similar in color to main object | yes, no |
| background color | red, blue, green, ... |
| is containing reflective surfaces | yes, no |
| is indoor | yes, no |
| weather | sunny, cloudy, rainy, ..., |
| time of day | morning, afternoon, evening, ... |
| **Global** | |
| **Attributes** | **Tags** |
| overall color temperature | warm, neutral, cool |
| image saturation | high, medium, low |
| resolution | high, medium, low |
| camera angle | level, high angle, low angle... |
| noise level | high, medium, low |
| brightness | high, medium, low |
| camera distance from main object | close-up, medium shot, wide shot |
| sharpness | sharp, medium, blurry |
| overall tone | warm, cool, neutral |
| image orientation | portrait, landscape |
| is blurred | yes, no |
| contrast | high, medium, low |

## A.5    DETAILED EXPERIMENTAL SETUP OF THE MODEL REPAIR

In the main paper, we select two model repair scenarios: data-augmented training (i.e., with the original training data) and model fine-tuning (i.e., without the original training data), across three different tasks to verify the model repair capability of HiBug2.

For image classification, the ResNet18 model we repair is initially trained on the first 500 images per class from ImageNet (Deng et al., 2009). We utilize different image search engines to obtain a new set of 5,000 images from the web, manually ensuring no overlap with the validation set used for debugging. We then select 250 images from this set to serve as the test set, while the remaining images are designated as the query set. Both HiBug2 and HiBug, along with random selection, choose a total of 500 images (100 images per class). For pose estimation, the query and test sets are derived from an industrial private dataset, with an overall setup similar to the image classification

task. For object detection, we employ RTMDet-X (Lyu et al., 2022) pretrained on the COCO dataset and fine-tuned on 1,000 images from the KITTI (Geiger et al., 2012) training set as the baseline model. We further select another 500 images from the KITTI (Geiger et al., 2012) training set for our experiment of model repair using the selection methods above and conduct a second-stage fine-tuning of the model. In this task, error slices are related to objects. For an image containing multiple objects, the average model performance regarding all objects are calculated and jointly determines the priority of the images in selection.

## A.6 USER STUDY

Table 6: User studies and comparisons between HiBug2 and the existing baseline HiBug

| Method | Coherence | Coverage | Utility | User Experience |
|---|---|---|---|---|
| HiBug2 (ours) | 4 | 4 | 4 | 4 |
| HiBug | 0 | 0 | 0 | 0 |

We conduct a user study to evaluate the model debugging capabilities of HiBug2 in comparison to HiBug (Chen et al., 2024). The study involves 4 participants, all of whom are machine learning and computer vision practitioners. Participants were presented with the methods, a classification model, and the dataset, and they used these methods to identify error slices. We quantify the user preferences based on four criteria: (1) slice coherence, (2) slice attribute coverage, (3) slice insight, and (4) overall user experience.

**Models and data.** The user study is conducted only on image classification tasks for better simplicity and objective clarity. We use a ResNet18 model with pretrained accuracy 71.2% on the bear species of ImageNet presented in the main paper.

**Metrics.** Given the dataset of bear species, the model, and the error slice discovery methods, the participants were asked to select a preferred method according to the following four criteria:

- **Slice coherence**, assessing whether the images within a slice exhibit consistent visual attributes and tags, and whether these attributes and tags accurately represent the selected data samples.
- **Slice attribute coverage**, evaluating whether the attributes captured in the slices comprehensively describe the majority of failure scenarios in the dataset.
- **Slice insight**, measuring the practical value of a slice for debugging and improving model performance. This includes evaluating the interpretability of a slice (i.e., whether the attributes and tags align with participants' expectations of failure scenarios) and its contribution to model repair (i.e., whether incorporating samples from the slice leads to potential performance improvements).
- **Overall user experience**, gauging user satisfaction with the UI design, clarity of the results, and system runtime.

**Results.** As shown in Table 6, all of the participants prefer HiBug2 across all metrics, showing significant improvements in debugging compared to HiBug.

## A.7 PSEUDO-CODE OF THE EFFICIENT SLICE ENUMERATION ALGORITHM AND BASELINES

We present the slice enumerations algorithms used in Section5.2 of the main paper. The naive algorithm (Algorithm 1) refers to the brute-force method mentioned in Section4.1.3. It simply lists all possible data slices and then searches for matching data for each slice. For the tree-structured baseline (Algorithm 2), compared with DebugAgent (Algorithm 3), this approach lacks the pruning and the intersection discussed in Section 4.1.4. Compared to the naive approach, the tree-structured baseline progressively increases the number of attributes included in each data slice during the search. For example, when searching for all possible data slices for a combination of three attributes, the algorithm first searches for slices with one attribute, then with two attributes, and finally with three attributes. When searching for matches for a three-attribute combination, the search is restricted to the matching data from the parent node, improving search efficiency.

---

**Algorithm 1** Naive Slice Enumeration

---

**Require:** Attribute set $A = \{a_1, a_2, \ldots, a_n\}$, tag sets for each attribute $T = \{T_1, T_2, \ldots, T_n\}$, tag annotations for all data $L$, data count threshold $M$, maximum attribute combination depth $D$

**Ensure:** Informative slice set $S$

  $S_{\text{full}} \leftarrow \texttt{GetAllCombinations}(A, T, D)$   ▷ *Generate all attribute-tag combinations up to depth D*

  $S \leftarrow \{\}$   ▷ *Initialize the final slice set*

  **for** each slice $s \in S_{\text{full}}$ **do**

    $s[\text{DATA}] \leftarrow \texttt{SearchMatchData}(L, s)$   ▷ *Find matching data for this slice*

    **if** $|s[\text{DATA}]| \geq M$ **then**

      $S \leftarrow S \cup \{s\}$   ▷ *Add valid slice to the final set*

    **end if**

  **end for**

  **return** $S$   ▷ *Return the informative slice set*

---

**Algorithm 2** Tree-Structured Slice Enumeration

---

**Require:** Attribute set $A = \{a_1, a_2, \ldots, a_n\}$, tag sets for each attribute $T = \{T_1, T_2, \ldots, T_n\}$, tag annotations for all data $L$, data count threshold $M$, maximum tree depth $D$

**Ensure:** Informative slice set $S$

  $S \leftarrow \{\}$   ▷ *Initialize the final slice set*

  $S_0 \leftarrow \{\}$   ▷ *Initialize the slice set for depth 0*

  **for** $d = 1$ **to** $D$ **do**

    $S_d \leftarrow \texttt{ExpandSlices}(S_{d-1}, A, T)$   ▷ *Generate slices by adding an attribute at each depth*

    **for** each slice $s \in S_d$ **do**

      $s[\text{DATA}] \leftarrow \texttt{SearchMatchData}(s[\text{PARENT}][\text{DATA}], L)$

    **end for**

  **end for**

  **for** $d = 1$ **to** $D$ **do**

    **for** each slice $s \in S_d$ **do**

      **if** $|s[\text{DATA}]| < M$ **then**

        Remove $s$ from $S_d$

      **end if**

    **end for**

    $S \leftarrow S \cup S_d$   ▷ *Add valid slices to the final set*

  **end for**

  **return** $S$   ▷ *Return the informative slice set*

---

**Algorithm 3** Efficient slice enumeration given the attribute and tag sets

---

**Require:** Attribute set $A = \{a_1, a_2, \ldots, a_n\}$, tag sets for each attribute $T = \{T_1, T_2, \ldots, T_n\}$, tag annotations for all data $L$, data count threshold $M$ for slice pruning, maximum tree depth $D$

**Ensure:** Informative slice set $S$

  $S \leftarrow \{\}$   ▷ *Initialize the final slice set*

  $S_0 \leftarrow \{\}$   ▷ *Initialize the slice set for depth 0*

  **for** $d = 1$ **to** $D$ **do**

    $S_d \leftarrow \texttt{MatchPairIntersection}(S_{d-1}, A, T)$

    **for** each slice $s \in S_d$ **do**

      $s[\text{DATA}] \leftarrow \texttt{SearchMatchData}(s[\text{PARENT}][\text{DATA}], L)$

      **if** $|s[\text{DATA}]| < M$ **then**

        Remove $s$ from $S_d$

      **end if**

    **end for**

    $S \leftarrow S \cup S_d$   ▷ *Add valid slices to the final set*

  **end for**

  **return** $S$   ▷ *Return the informative slice set*

---

## A.8    Prompts in HiBug2

For better understanding HiBug2, we provides three important prompts in HiBug2. The few-shot examples are omitted. For all the prompts in HiBug2, please refer to our code.

**Extracting attributes by a comparative approach.** This approach is introduced in Section 3.2.1, it extracts attributes that varied in images of a dataset.

> *You are a dedicated assistant for spotting the differences between two images and summarizing them into common visual attributes.*
>
> *1. Inputs Provided:*
>
> *- Two images featuring the same main object class.*
>
> *- The class of main objects.*
>
> *- A JSON form with keys: "main object", "background", and "global". Each key contains a list of visual attributes.*
>
> *2. Tasks:*
>
> *- Analyze the visual differences between the two images and propose new visual attributes that highlight these differences.*
>
> *- Add these new attributes to the corresponding list in the JSON form:*
>
> *1."main object": Attributes related to the main object itself (e.g., "object color", "object size", "object clothes").*
>
> *2."background": Attributes related to the background scene (e.g., "background color", "is sky presented", "natural habitat").*
>
> *3."global": Attributes related to the overall image quality (e.g., "brightness", "contrast").*
>
> *- When you refer the main object class name in attributes, such as "teddy bear color", write it as "object color".*
>
> *- Ensure that each attribute is concise, specific, and clearly describes a visual feature relevant to the main object class and the category. For example, "object color" is valid, but "overall appearance" is too vague.*
>
> *- Avoid generating attributes that overlap significantly with each other. Each attribute should describe a distinct feature.*
>
> *- If an attribute's value is expected to be "yes" or "no", prepend the attribute name with "is ". For example, "trees presence" should be written as "is trees presented".*
>
> *- Attribute names should clearly reflect the type of value that should be filled in, when an image is given. Avoid vague or general names. e.g., if the difference is whether trees in the image, use "is trees presented" instead of just "trees". If the difference is the color of trees, use "trees color".*
>
> *- Visual attributes should be concise, specific, and clearly describe a visual feature. For example, "object background" and "overall appearance" is too vague.*
>
> *- Only propose attributes that clearly differentiate the two images. Avoid generating redundant or insignificant attributes.*
>
> *3. Outputs:*
>
> *- Provide the updated JSON object with keys "main object", "background", and "global", each containing a list of visual attributes. Ensure the attributes are unique and relevant.*
>
> *- Your output should include the form only.*

**Generating an initial list of potential and unbiased tags.** This approach is introduced in Section 3.2.2, it establish the semantic scope and granularity for tags of each attribute.

> *You are a dedicated assistant for finding all possible tags corresponding to specific visual attributes in images.*
>
> *Each time, the users will provide you with the following information:*
>
> *- The main object of the images.*
>
> *- A JSON form with keys "main object," "background," and "global," where each value is a list of visual attributes. Each visual attribute belongs to its corresponding key.*
>
> *Your task is to list all possible tags for each visual attribute and output a JSON form:*
>
> *1. Categories:*

- Attributes under "main object" relate specifically to the given main object, attributes under "background" relate to the background scene, and attributes under "global" relate to the overall image quality.

2. Tag Generation:

*Basic Tag List Creation*:

- For each visual attribute, provide a comprehensive list of possible tags. Each tag should represent a distinct and commonly observable feature related to that attribute.

- Tags should be short, concise, and easily understandable.

*Handling Different Attributes*:

- For attribute start with "is ", such as "is tree presented". The tag list is ["yes", "no"].

- For attribute not start with "is ", the tag list enumerates the possible situations or categories, rather than a yes/no judgment.

*Categorization for Vast Tag Options*:

- If an attribute has a vast or infinite number of possible tags (e.g., "background types"), categorize these tags into meaningful groups and list the category names (e.g., "indoor," "outdoor").

*Contextual Appropriateness*:

- Ensure the tags are contextually appropriate for the main object. For instance, if the main object is "fish," the attribute "background type" should include tags like "coral reef," but not "sky."

*Avoiding Redundancy and Overlap*:

- Avoid redundancy in tags. Each tag should be unique within its list and clearly distinguishable from others (e.g., avoid both "big" and "large" in the same list).

- Avoid generating attributes that overlap significantly with others. e.g., for attribute "object color", avoid both specific colors (like "red" or "blue") and "multicolor" in the same list.

*Completeness of Tag Lists*:

- Aim to make the list of tags as complete as possible. Any commonly seen image of the main object should be able to match at least one tag for each attribute.

- Propose at least two tags for each attribute, ensuring a variety of relevant options.

*Handling Exceptions*:

- The main object appear at least partially in the image, while the presence of background objects may vary.

- If an attribute pertains to a background object or a specific component of the main object, include "not visible" in the tag list to account for cases where that element might not be visible.

3. Outputs:

- Present the results in the same JSON format as the input. The output should be a dictionary with keys "main object," "background," and "global," where the values are dictionaries. In these dictionaries, each visual attribute name is a key, and the corresponding value is a list of tags.

- Keep the name of attributes from the input form unchanged. For example, if the name of an attribute is "object size", you should keep its name "object size" unchanged in the output form.

- Your output should include the JSON form only.

**Predicting unseen error slices for classification task.** This approach is introduced in Section 4.2, it instructs GPT to predict potential error slices that blur boundaries between closely related categories. It is used for classification task only.

You are a dedicated assistant for predicting attribute-tag combinations that will make data from one class resemble another class, thereby confusing existing image classification neural network models.

Each time, the users will provide you with the following information:

- The class of the main object.

- The target class for confusion.

- A json form that records all attributes and tags involved. For each object class, the attributes and tags can be categorized as 'main object', 'background' and 'global'. Each attribute corresponds to multiple tags. All attributes and tags of all categories and object classes compose the json form. - Visual attributes in "main object" are related to the given main object. Visual attributes in "background" are related to the background scene. Visual attributes in "global" are related to the image quality.

*- A positive integer.*

*Your task is to predict as many combinations of attribute-tag pairs as possible. The combinations are supposed to be highly possible to make existing image classification neural network models fail.*

*- Your output is a form. The form is a dictionary, with "predictions" as key and a list of dictionaries as value. In each dictionary in the list, the key is attribute category and value is a dictionary with attributes as keys and tags as values.*

*- You need to predict as many combinations as possible.*

*- You need to use the given attributes and tags, and not create new ones.*

*- For each predicted attribute, you need to assign one and only one tag.*

*- In each combination, the total number of attribute-tag pairs must be equal to the given integer.*

*- In the predicted combinations, there can be multiple attributes of the same category.*

*- You need to consider the class of the main object and the target class for confusion. You should ensure that the predicted combinations are highly possible to make existing models confuse main object class with target object class. For example, the predicted combinations might make the main object looks like the target class.*

*- You output the form only. No explanation in your output.*

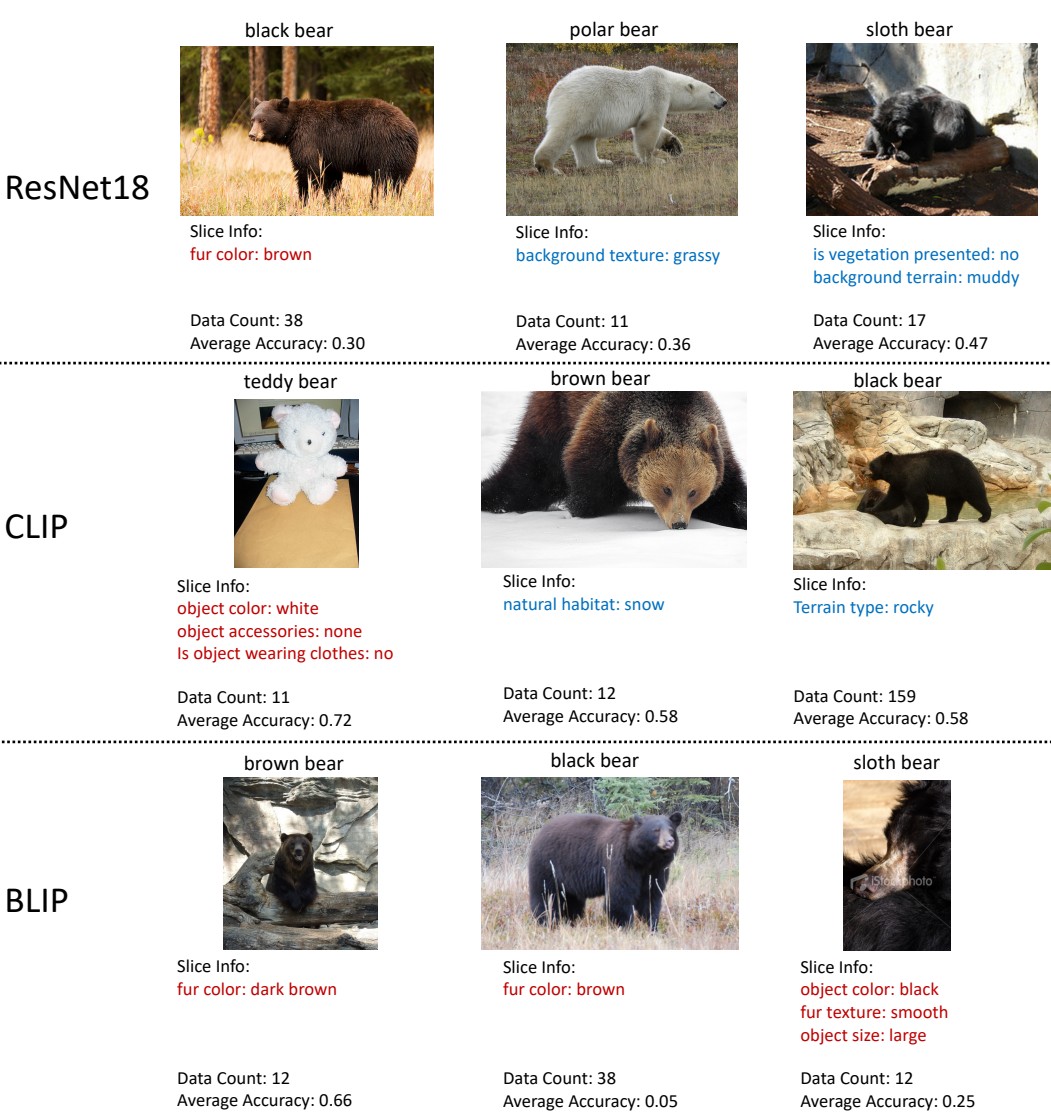

Figure 8: Identified slices of the image classification task by HiBug2.

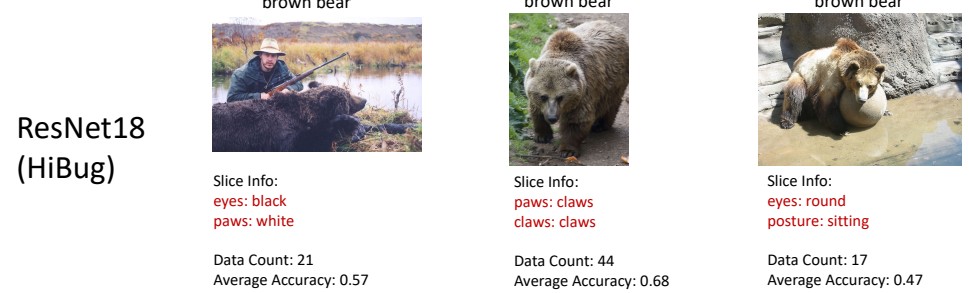

Figure 9: Identified slices of the classification task by baseline method HiBug.

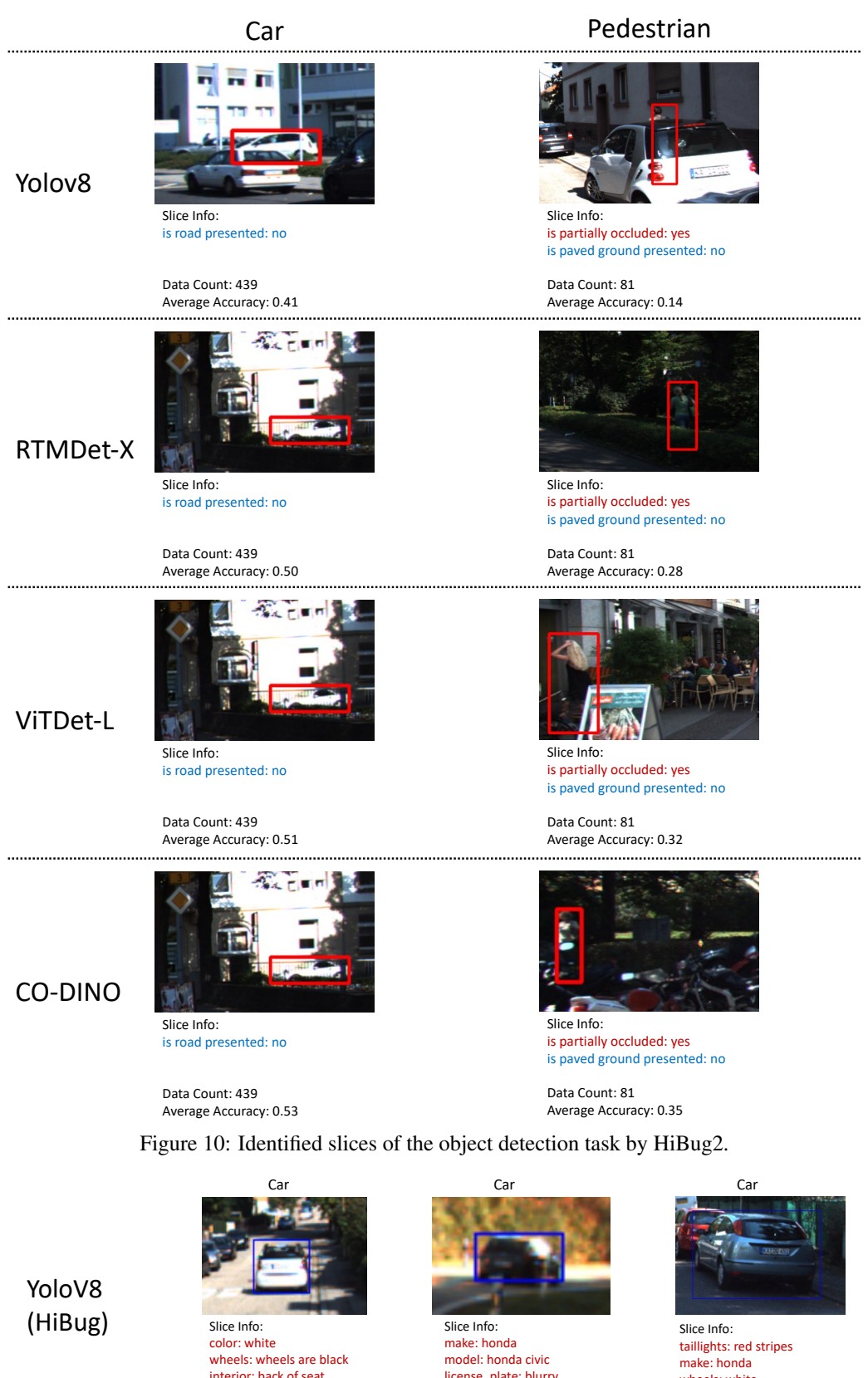

Figure 10: Identified slices of the object detection task by HiBug2.

Figure 11: Identified slices of the object detection task by baseline method HiBug.

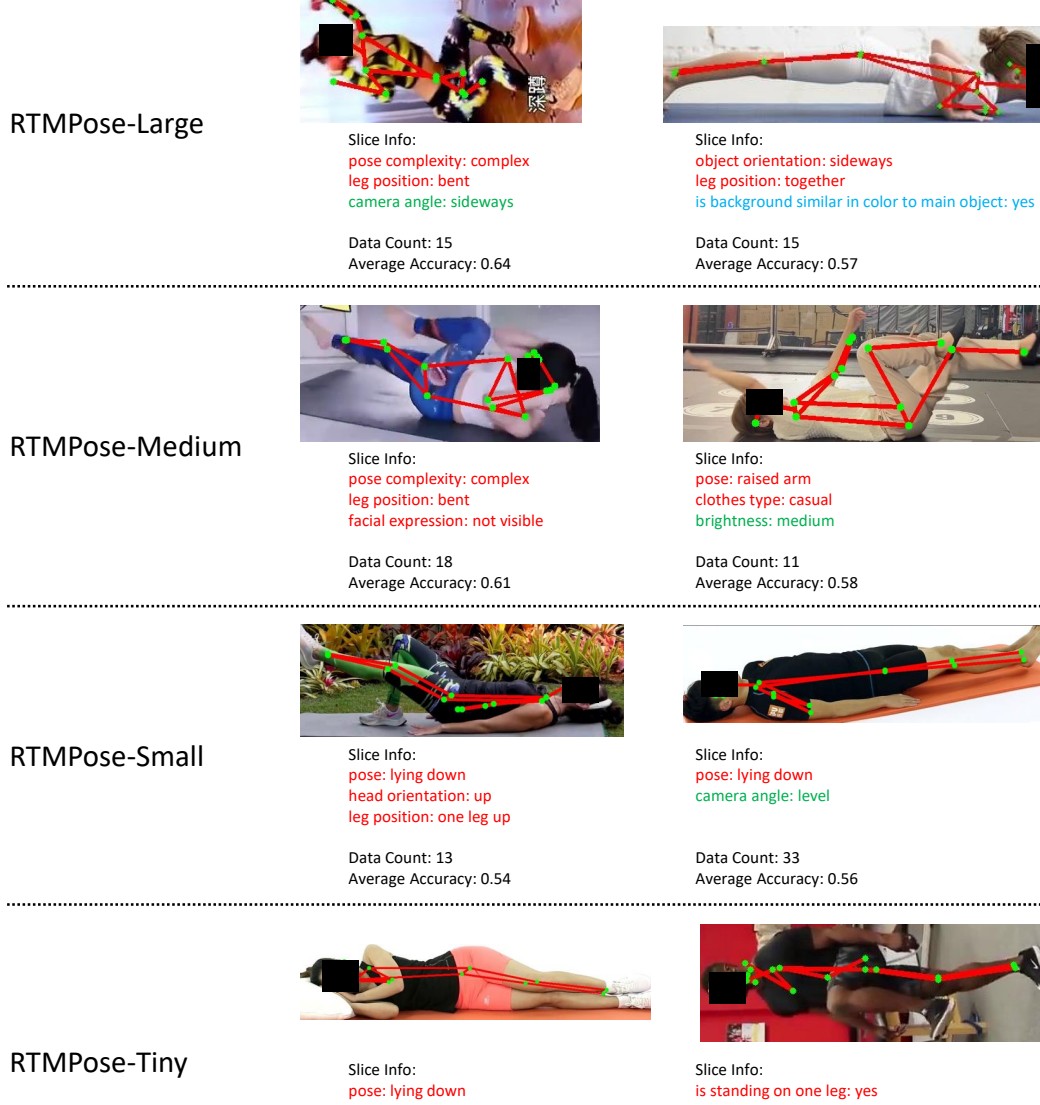

Figure 12: Identified slices of the pose estimation task by HiBug2.

