# OpenReview forum: "HiBug2: Efficient and Interpretable Error Slice Discovery for Comprehensive Model Debugging"
_ICLR.cc/2025/Conference — ICLR 2025 Poster_

### Official Review · Reviewer_2HXz · 2024-11-02

**Soundness:** 3
**Presentation:** 3
**Contribution:** 3
**Rating:** 6
**Confidence:** 3

**Summary:**

This paper proposes a method called DebugAgent for the automated discovery of error slices in deep learning models and for model repair. The approach first uses GPT to generate attributes for the dataset and to assign tags to the data. Then, based on the generated attribute-tag pairs, it employs tree-structured enumeration for clustering and identifying error slices. Finally, the method utilizes these error slices to repair the model. To validate the effectiveness of DebugAgent, the authors compared it with previous methods in terms of the generated attributes and tags, clustering efficiency, and model repair effectiveness.

**Strengths:**

1. The focus of the paper on error slice discovery is important, as understanding these error slices is crucial for comprehending the incorrect behavior of deep learning models. The paper presents an automated method aimed at addressing this issue of error slice discovery and model repair.

2. The proposed method's effectiveness was validated across multiple task scenarios (image classification, pose estimation, and object detection), demonstrating improvements in the quality of attribute and tag generation, as well as model repair effectiveness compared to previous methods.

**Weaknesses:**

1. DebugAgent employs GPT in several parts of the algorithm, including attribute and tag generation, as well as instruction-based slice generation. However, the specific prompts used in these processes are not detailed in the paper. It would be beneficial to provide the prompts used in these steps to enhance the understanding of the method.

2. In the model repair experiment section, the improvements of DebugAgent over baseline methods are not significant. Given the randomness inherent in the training process of deep learning models, it would be beneficial to clarify whether these results are based on repeated experiments. If not, it is necessary to conduct repeated experiments to mitigate the effects of randomness.

3. DebugAgent identifies different error slices through clustering. It would be beneficial to verify whether the error slices from different clusters correspond to distinct model bugs. For instance, one could repair the model using the error slices from a specific cluster and then evaluate the model's performance on that cluster compared to its performance on other clusters to assess any differences.

**Questions:**

1. Can the authors provide the prompts used in the method?

2. Were repeated experiments conducted in the experimental section to mitigate the impact of randomness on model repair?

3. Is it possible to verify whether the error slices from different clusters correspond to distinct bugs in the model?

---

> ### Author Response · Authors · 2024-11-17
>
> ### Q1. Can the authors provide the prompts used in the method?
>
> Thank you for the suggestion. We have added an additional section (A.8) in the appendix of the revised version to provide some prompts used in our method. Please note, we have already provided most of the prompts in the Supplementary Material. We will open-source all the code and framework if the paper gets accepted, and the prompts will be included as well.
>
> ### Q2. Were repeated experiments conducted in the experimental section to mitigate the impact of randomness on model repair?
>
> Thank you for the reminder. In the revised version, we have updated Table 2 with the average results from five repeated runs using different random seeds. The overall performance remains largely consistent across these runs.
>
> | Method  | Image Classification | Pose Estimation | Object Detection |
> | --- | --- | --- | --- |
> | DebugAgent (ours)  | 0.839 (+7.6%)  | 0.839 (+2.1%) | 0.683(+4.9%) |
> | HiBug | 0.832 (+6.3%) | 0.831 (+1.1%) | 0.673 (+3.4%) |
> | Random | 0.817 (+4.7%) | 0.829 (+0.9%) | 0.655 (+0.6%) |
> | Original | 0.780 | 0.822 | 0.651 |
>
> ### Q3. **Is it possible to verify whether the error slices from different clusters correspond to distinct bugs in the model?**
>
> Thank you for this insightful question; it addresses an important aspect of error slice discovery and model debugging. We have added this interesting experiment in the appendix, section A.1.6, of the revised version.
>
> In our paper, error slices are defined based on the combination of tags, and therefore, slices may overlap in some tags. While error slices are distinct in terms of their tag combinations, they do not necessarily correspond to distinct bugs. Fixing one error slice may improve the model’s performance on other related error slices as well.
>
> To test this, we designed a simple experiment to check the impact of fixing one error slice on others. For the ResNet model used in the classification task described in the main paper, we chose an error slice corresponding to class teddy bear, defined as (object color: white, object pose: sitting). We then queried the data matching this error slice (20 images) from a hold-out dataset. We fine-tuned the model on this data for 20 epochs and evaluated its accuracy on the validation set. We collected the model’s accuracy on three types of data:
>
> 1. Data matching the selected error slice.
> 2. Data belonging to error slices that overlap with the selected slice (e.g., (object color: white, xxx), (xxx, object pose: sitting)).
> 3. Data belonging to error slices that does not overlap with the selected slice.
>
> |  | Match | Overlap | Non-overlap |
> | --- | --- | --- | --- |
> | **Before** | 0.500 | 0.840 | 0.829 |
> | **After** | 0.875 | 0.862 | 0.791 |
>
> The results reveal a significant improvement in the model’s performance on the fixed error slice. Interestingly, performance on overlapping error slices also improved, suggesting a potential relationship between these slices and shared model weaknesses. However, performance on non-overlapping data decreased slightly, likely due to overfitting to the specific distribution of the fine-tuning data. This experiment suggests that while error slices are defined by distinct tag combinations, they are not distinct bugs.

---

> > ### Comment · Reviewer_2HXz · 2024-11-23
> >
> > Thank you for answering my question. I will keep my score.

---

> > > ### Author Response · Authors · 2024-11-29
> > >
> > > Dear Reviewer 2HXz
> > >
> > > Thank you for your detailed and constructive feedback, which has greatly helped us improve our paper. As we approach the end of the discussion phase, we would like to kindly ask if there are any remaining concerns or aspects where you feel could further improve our paper.
> > >
> > > Thank you once again for your valuable time and feedback.
> > >
> > > Sincerely,
> > >
> > > Authors

---

### Official Review · Reviewer_KQpB · 2024-11-04

**Soundness:** 3
**Presentation:** 2
**Contribution:** 2
**Rating:** 6
**Confidence:** 3

**Summary:**

An entirely automated closed-loop debugging framework, DebugAgent, has been introduced for error slicing discovery and model rectification. DebugAgent encompasses attribute and label generation, error slicing discovery, and model rectification. Besides, a structured generation process has been implemented to create comprehensive visual attributes. An efficient slicing enumeration algorithm has been developed leveraging the unique features of data slices to alleviate the issue of combinatorial explosion. This paper employs feature-based label replacement and instruction-based approaches to address errors invisible outside the validation set.

**Strengths:**

Originality: The method for generating attributes and labels takes into account environmental factors and image quality metrics, enabling the capture of nuanced details that are highly pertinent to erroneous slice discovery. The slice enumeration approach effectively expedites the enumeration process.
Quality: Good. Further elucidation and interpretation of the experimental results, such as those in Section 5.4, is needed.
Clarity: Fair. However, some procedures still lack clarity, and the comparative experiments are not sufficiently comprehensive.
Significance: The approach has enhanced the consistency and precision in identifying erroneously sliced samples, while significantly bolstering the model's reparative capabilities.

**Weaknesses:**

1、The formatting of Figures 1 and 4 lacks aesthetic appeal; for Tables 1 and 2, it is recommended to utilize triple-line tables.
2、In the experimental section, the efficacy of attribute and label generation, as well as slice enumeration, was individually assessed in Part V, without comprehensive comparisons to existing methodologies.
3、In the abstract, it is mentioned that DebugAgent visualizes the attributes of task generation through an interpretable and structured process. While this structured approach is referenced elsewhere in the text, its specific meaning in the context of utilizing GPT for attribute and label generation in Section 3.2 may not be entirely clear.
4、In Section 4.2, the mention of tag substitution and instruction-based method is aimed at predicting erroneous segments beyond the validation set. While Table 1 illustrates model degradation in predicting erroneous slices using these methods, Section 5.4 reveals that the model's average performance on these predicted erroneous slices significantly falls below its overall performance, emphasizing the efficacy of the DebugAgent approach in predicting additional erroneous segments. Your query highlights a potential inconsistency: the absence of testing the effectiveness of the DebugAgent approach in predicting additional erroneous segments, coupled with the fact that tag substitution and instruction-based method are proposed by your team, revealing a logical gap in the narrative.
5、Some data points are referenced in Section 5.3 without accompanying tables for display.

**Questions:**

1. Table 1 presents the degradation of the model's performance in predicting erroneous segments when using label replacement and instruction-based approaches. Negative values in the table indicate a decrease in prediction accuracy compared to the baseline. Regarding the relationship between the data results and conclusions in Section 5.4, the section shows that the model's average performance on predicted erroneous segments is notably lower than its overall performance. This emphasizes the effectiveness of the DebugAgent approach in predicting additional erroneous segments. However, your concern is valid: while the paper introduces label replacement and instruction-based methods as supplements to the DebugAgent approach, Section 5.4 only validates the effectiveness of these internally proposed methods. The absence of comparative analyses and the negative results presented raise questions about the justification of the proposed methods' effectiveness in the absence of further comparisons.

2、Comparative analysis with existing methods for comprehensive erroneous slice discovery is indeed a crucial aspect in research. It appears that the experiment comparing the proposed methods with existing approaches for erroneous slice discovery is missing in the paper, yet such comparisons play a significant role in assessing the effectiveness and advancements of the proposed techniques.

---

> ### Author Response · Authors · 2024-11-17
>
> ### W1. The formatting of Figures 1 and 4 lacks aesthetic appeal; for Tables 1 and 2, it is recommended to utilize triple-line tables.
>
> Thank you for the suggestion. In the revised version, we have updated Tables 1 and 2 to use triple-line tables for improved clarity and presentation. We have also updated Figure 4 to enhance its aesthetic appeal. For Figure 1, its current design prioritizes simplicity and clarity for conveying the information effectively. Adding additional elements, such as colors, may compromise its readability and detract from its intended purpose.
>
> ### W2.  In the experimental section, the efficacy of attribute and label generation, as well as slice enumeration, was individually assessed in Part V, without comprehensive comparisons to existing methodologies.
>
> Thank you for your comment. We would like to clarify that we have compared the efficacy of attribute and label generation to existing methods, as shown in section 5.1 and appendix A.2.
>
> As for slice enumeration, since the tag-then-slice approach is a relatively recent development, this issue has not been specifically addressed by previous methods. Specifically, HiBug focuses on error slice discovery based on a single attribute tag and does not consider attribute combinations. However, we found a similar slice discovery algorithm in HiBug's code, which has a structure similar to the naive approach used in our baseline method. Given the limited number of existing methods for comparison, we only include comparisons with custom-built baseline algorithms.
>
> ### W3. In the abstract, it is mentioned that DebugAgent visualizes the attributes of task generation through an interpretable and structured process. While this structured approach is referenced elsewhere in the text, its specific meaning in the context of utilizing GPT for attribute and label generation in Section 3.2 may not be entirely clear.
>
> Thank you for your comment. We understand that the term "visualizes" in the review may have been misunderstood. In the abstract, we state that DebugAgent generates task-specific visual attributes through an "interpretable and structured process."
>
> The "interpretable" aspect refers to two key points: first, our approach follows a clear and logical structure; and second, the generated attributes and tags are human-understandable. The "structured" aspect refers to our systematic method for prompting GPT to generate relevant attributes. Specifically, we first define three types of visual attributes and two sources of error. Then, we extract the visual attributes based on the intrinsic difficulty of the task and the characteristics of the dataset.
>
> ### W4. In Section 4.2, the mention of tag substitution and instruction-based method is aimed at predicting erroneous segments beyond the validation set. While Table 1 illustrates model degradation in predicting erroneous slices using these methods, Section 5.4 reveals that the model's average performance on these predicted erroneous slices significantly falls below its overall performance, emphasizing the efficacy of the DebugAgent approach in predicting additional erroneous segments. Your query highlights a potential inconsistency: the absence of testing the effectiveness of the DebugAgent approach in predicting additional erroneous segments, coupled with the fact that tag substitution and instruction-based method are proposed by your team, revealing a logical gap in the narrative.
>
> If we understand this comment correctly, the concern is the lack of baselines for evaluating our methods of predicting unseen slices. We clarify that:
>
> 1. Predicting erroneous slices is a novel aspect of our work that rarely attempted in previous error slice discovery methods, which is why there is no direct baseline comparison.
> 2. The primary function of DebugAgent is to systematically identify and repair erroneous slices based on the dataset validation set. In contrast, tag substitution and instruction-based methods are auxiliary tasks used to hypothesize additional erroneous slices that may not be covered by the validation set.
>
> We have clarified this distinction in the revised version to emphasize that the lack of baseline comparison stems from the novelty of our approach.
>
> ### W5. Some data points are referenced in Section 5.3 without accompanying tables for display.
>
> Thank you for your comment. The data points referenced in Section 5.3 refer to the number of error slices identified for each dataset and model. However, since the number of models used varies across datasets, it would be difficult to present this information in a single, unified table.

---

> > ### Author Response · Authors · 2024-11-17
> >
> > ### Q1.
> >
> > Please refer to our answer to W4.
> >
> > ### Q2. Comparative analysis with existing methods for comprehensive erroneous slice discovery is indeed a crucial aspect in research. It appears that the experiment comparing the proposed methods with existing approaches for erroneous slice discovery is missing in the paper, yet such comparisons play a significant role in assessing the effectiveness and advancements of the proposed techniques.
> >
> > Thank you for the insightful question. Comparing error slices is inherently challenging due to their subjective nature, as aspects such as coherence often rely heavily on user perception. To address this subjectivity, we conducted a user study, detailed in Appendix A.6, where feedback was gathered to evaluate error slices generated by different methods.
> >
> > The quality of error slices is closely tied to the attributes and tags used to define them, which can vary significantly between methods. For instance, HiBug's attributes primarily focus on the main object, and its tags for certain attributes, such as 'eyes', show inconsistencies (e.g., 'round' for shape and 'black' for color). These limitations can negatively affect both the coherence of the error slices and the user experience in practical applications. Comparisons of attributes and tags are provided in Section 5.1 and Appendix A.3. Additionally, to further compare error slices, we have included visualizations and comparisons of error slices discovered by our method and baselines in Appendix A.4 of the revised version.

---

> ### Author Response · Authors · 2024-11-27
>
> Dear Reviewer KQpB,
>
> Thank you again for your feedback and for engaging with our work. We wanted to kindly follow up regarding the discussion. While the discussion period remains open for another week, tomorrow marks the final deadline for any further revisions to the paper.
>
> If there are any additional concerns, experiments, or clarifications that you feel are necessary, we would be happy to address them promptly within the discussion. Please let us know if there is anything we can do to further improve the paper or clarify our responses.
>
> Thank you once again for your time and effort.

---

> > ### Comment · Reviewer_KQpB · 2024-11-30
> >
> > Thank you for your response. I raised my score.

---

### Official Review · Reviewer_3dP8 · 2024-11-04

**Soundness:** 2
**Presentation:** 4
**Contribution:** 2
**Rating:** 6
**Confidence:** 4

**Summary:**

The article presented DebugAgent, an approach for error slice discovery for image classification, pose estimation, and
object detection. The approach consists of three main blocks: attribute and tag generation, error slice discovery, and
model repair. In the experiments, they evaluate the different components of their approach, such as the slice enumeration algorithm, to address the challenges of the combinatorial explosion during the slice exploration and conclude by comparing their model repair against HiBug, another automated approach. The work presents performance improvement upon the SOTA, although the time complexity comparison is done only between baseline versions of the author's method.

**Strengths:**

## S1 - Organization
The paper is unambiguous, well-structured, and well-written.

## S2 - Relevance
The problem is interesting and relevant to the community.

## S3 - Approach
The approach is well-articulated, simple to understand, and easy to implement.

## S4 - Coverage
The slice discover problem comprises several challenges in computer vision. The paper effectively presents and addresses them, and the experiments demonstrate the potential of DebugAgent in all of them.

**Weaknesses:**

## W1 - Baselines (Main issue)
The work lacks experiments with a broader set of SOTA methods; the authors comment on the tags generated by two additional methods, Domino and HiBug, but do not use them for most of the quantitative comparison. The complexity analysis tests the technique against a not-detailed naive approach and a baseline version of one of DebugAgent's algorithms. DebugAgent shows marginal improvement over the HiBug method when testing the performance improvement in model repair, but the reader does not know how costly this method is. The other experiments show that DebugAgent can identify salient slices, but nothing is said about SOTA. The only discussion on the matter is that different methods employ varied workflows, making comparing them difficult, but the authors do not specify the challenges that need to be overcome in order to make the other methods comparable (Please see questions Q3 and Q4).

## W2 - Scalability
Using more instance types, for example, the other 7 classes of the KITTI dataset, in the experiments would better understand the method's generalization capabilities. The authors discuss the scalability regarding tag generation in the appendices, but in the main text, there is only a time analysis of the slice enumeration task. How does the tag generation task scale with the data load? (Please also refer to Q3)

## W3 - Reproducibility
Including private datasets limits the reproducibility of the research; an indication of a public proxy dataset of a description of the pose estimation dataset's metadata would improve the reasoning of the comparison challenge (please see question Q5). There is also no specification for the baseline setups. Is it possible to have a detailed description of the naive approach and the baseline tree-structured method in the appendices? (Please see Q1 and Q2)

## W4 - Integration
A deeper ablation study would help understand the proposed method and how all algorithms are integrated. The ablation mentioned in Figure 5 can be considered just part of the time analysis of the slice enumeration task.

**Questions:**

- Q1 - What is the naive approach to slice enumeration?
- Q2 - How is the tree-structured baseline approach different from DebugAgent's method?
- Q3 - How difficult is comparing tag generation time between DebugAgent and the SoTA? (At least HiBug).
- Q4 - Is it possible to identify slices from other SoTA tag generators and compute their performances with the same models?
- Q5 - Is it possible to provide the shape (number of instances, features, and unique labels) of the private dataset used for pose estimation or an alternative proxy public dataset?

---

> ### Author Response · Authors · 2024-11-17
>
> ### Q1: What is the naive approach to slice enumeration?
>
> Thank you for pointing this out. This algorithm refers to the brute-force method mentioned in Section 4.1.2. It simply lists all possible data slices and then searches for matching data for each slice. We have now added this explanation in the revised version of the main text (section 5.2) and included an algorithm description in the appendix, section A.7, for further clarification.
>
> ### Q2: How is the tree-structured baseline approach different from DebugAgent's method?
>
> Thank you for raising this point. Compared with DebugAgent, this approach lacks the pruning and the Intersection discussed in Section 4.1.4. Compared to the naive approach, the tree-structured baseline method progressively increases the number of attributes included in each data slice during the search. For example, when searching for all possible data slices for a combination of three attributes, the algorithm first searches for slices with one attribute, then with two attributes, and finally with three attributes. When searching for matches for a three-attribute combination, the search is restricted to the matching data from the parent node, improving search efficiency. We have now added this explanation in the revised version of the main text (section 5.2) and included an algorithm description in the appendix, section A.7.
>
> ### Q3: How difficult is comparing tag generation time between DebugAgent and the SoTA? (At least HiBug).
>
> Thank you for your insightful question. Comparing the tag generation time between DebugAgent and SoTA methods such as HiBug is challenging due to the different factors that influence time consumption. For HiBug, the primary factor is GPU capacity, while for DebugAgent, it is network bandwidth.
>
> The most time-consuming part of attribute and tag generation (also of the whole method) is the tag assignment for all the images in the dataset, which typically accounts for over 95% of the total time, with larger datasets increasing this proportion. HiBug uses the BLIP VQA model, whereas our method uses GPT-4V. In practice, for HiBug, generating labels for a single attribute on one image takes around 2 seconds (using a 4060 GPU), and generating labels for all attributes for that image takes about 2×M seconds (where M is the number of attributes). For DebugAgent, generating labels for all attributes for one image takes around 6 seconds (using the GPT-4V API). If we consider generating labels for each image individually, HiBug’s time complexity is O(N×M), where N is the total number of images and M is the number of attributes. Our method, however, operates with O(N) because GPT-4V can generate all attributes in one step.
>
> In practice, parallel processing is typically used to accelerate these processes. With unlimited GPUs, HiBug could label all data simultaneously, and similarly, with unlimited threads and network bandwidth, DebugAgent could generate labels for all data in 6 seconds. Due to the different influencing factors, it is difficult to establish a fair and standardized comparison.
>
> Thanks again for pointing out this issue. Time cost of DebugAgent is indeed an aspect that we did not detail in the paper. We have now added a discussion of the time cost of DebugAgent to the revised version of appendix, section A.1.5.
>
> ### Q4:  Is it possible to identify slices from other SoTA tag generators and compute their performances with the same models?
>
> Yes, it is possible. Our slice enumeration algorithm can be directly applied to HiBug. Additionally, our method can be used on datasets that already have attributes and tags, such as the CelebA dataset, as long as all images in the dataset are labeled with a consistent set of attributes. However, our method cannot be applied to general tag generators, such as Recognize Anything[1], because these methods do not label images with a consistent set of attributes, which hinders the implementation of slice enumeration.
>
> ### Q5: Is it possible to provide the shape (number of instances, features, and unique labels) of the private dataset used for pose estimation or an alternative proxy public dataset?
>
> The pose dataset contains 47,057 images (24,832 images meticulously curated from COCO, the remainder from a private source), primarily used for rehabilitation training in hospitals to recognize patient movements and assess whether the designed exercises meet the required standards. In our experiments, models are pre-trained on the COCO portion. For error slice discovery, we use 7,057 images from the private portion as the validation set. For model repair, we select 1,241 images from the private portion. We have included these information in the appendix, section A.3, of the revised version. We have also presented examples of error slices from this dataset (with faces blurred for privacy) in the appendix, section A.4, of the revised version.

---

> > ### Author Response · Authors · 2024-11-17
> >
> > ### Additional discussion: the difficulty of comparisons.
> >
> > Inspired by reviewer’s insightful comments, we would like to further discuss why the comparison with other error slice discovery methods is difficult. We have also added these discussion in the appendix, section A.1.6, of revised version.
> >
> > Unlike other fields, we cannot find a standardized evaluation process for error slice discovery methods. Most studies design custom-built datasets and conduct limited comparisons. We identify two key challenges:
> >
> > 1.  Differences in workflow. Even when methods aim for similar goals, as mentioned in our related work section, some methods rely heavily on human analysis with LLM-assisted slice discovery, while others cluster data before human labeling. Approaches like ours and HiBug, however, label the data first and then discover slices based on attribute clustering. These workflow differences make it challenging to establish suitable baselines for evaluating each component of the method.
> > 2. Error slices do not have ground truths. Error slices require shared human-understandable attributes, but even for the same group of data, different individuals may define and label attributes in vastly different ways [2].
> >
> > Due to these difficulties, we only compare with few baselines. For example, in the evaluation of slice enumeration experiments, there are no suitable baseline methods with the same task objective for comparison. While HiBug follows a similar process that labels the data before performing slice enumeration. It focuses on slice enumeration based on a single attribute tag and does not consider attribute combinations. We found a slice discovery algorithm in HiBug’s code, which aligns with the naive method used in our baseline to support combination enumeration.
> >
> > [1]: Zhang Y, Huang X, Ma J, et al. Recognize anything: A strong image tagging model[C]//Proceedings of the IEEE/CVF Conference on Computer Vision and Pattern Recognition. 2024: 1724-1732.
> >
> > [2]: Johnson N, Cabrera Á A, Plumb G, et al. Where does my model underperform? a human evaluation of slice discovery algorithms[C]//Proceedings of the AAAI Conference on Human Computation and Crowdsourcing. 2023, 11(1): 65-76.

---

> > > ### Comment · Reviewer_3dP8 · 2024-11-22
> > >
> > > Thank you for your answers to our questions and for your clarifications, in particular about the comparison baselines. I remain positive about the work and keep my score.

---

> > > > ### Author Response · Authors · 2024-11-29
> > > >
> > > > Dear Reviewer 3dP8
> > > >
> > > > Thank you for your detailed and constructive feedback, which has greatly helped us improve our paper.  As we approach the end of the discussion phase, we would like to kindly ask if there are any remaining concerns or aspects where you feel could further improve our paper.
> > > >
> > > > Thank you once again for your valuable time and feedback.
> > > >
> > > > Sincerely,
> > > >
> > > > Authors

---

### Author Response · Authors · 2024-11-17

We are grateful for the thoughtful feedback from the reviewers and their recognition of the key strengths of our paper. Reviewer 3dP8 praised the clarity, structure, and relevance of our work, while Reviewer KQpB highlighted the originality and efficiency of our method in attribute generation and slice enumeration. Reviewer 2HXz commended the importance of focusing on error slice discovery and the demonstrated effectiveness of DebugAgent across multiple tasks. These acknowledgments validate the significance and impact of our contributions.

In the revised version, we have made several updates to improve the clarity, completeness, and aesthetic appeal of the paper:

- **Tables and Figures**: We updated Table 1, Table 2, and Figure 4 to enhance their aesthetic presentation. Additionally, the values in Table 2 have been updated with results averaged over five random seeds.
- **Baselines**: Section 5.2 and Appendix A.7 have been revised to provide a clearer introduction to the baseline methods used in the evaluation of the slice enumeration algorithm.
- **Time Cost Discussion**: We added a detailed discussion about the methods' time cost in Appendix A.1.5.
- **Comparison Challenges**: A new discussion on the inherent difficulty in comparing error slice discovery methods has been included in Appendix A.1.6.
- **Cluster Analysis Experiment**: An additional experiment exploring whether error slices from different clusters correspond to distinct model bugs has been added in Appendix A.1.7.
- **Dataset Descriptions**: Extra details about the datasets are now included in Appendix A.3.
- **Error Slice Visualization and Comparison**: We provided more visualizations of error slices and comparisons with those generated by baseline methods in Appendix A.4.
- **Prompts**: Examples of prompts used in our method have been added in Appendix A.8.

These revisions directly address the reviewers’ suggestions and concerns, providing a more comprehensive and detailed presentation of our work.

We also noticed a shared concern from reviewers 3dP8 and KQpB regarding the limited number of compared baselines. This challenge arises from the diversity in workflows among existing error slice discovery methods. Some methods rely heavily on human analysis with LLM-assisted slice discovery, while others cluster data before human labeling. Approaches like ours and HiBug, however, label the data first and then discover slices based on attribute clustering. Consequently, it challenging to establish suitable baselines for evaluating each component of the method.

For the evaluation of the slice enumeration algorithm, we cannot find suitable baseline methods with the same task objective for direct comparison. HiBug, which follows a similar process of labeling data before performing slice enumeration, focuses on slices based on single attribute tags and does not consider attribute combinations. The slice discovery algorithm found in HiBug’s code is aligned with the naive method in our baseline.

Regarding the prediction of unseen slices, this aspect is a novel contribution of our work and has rarely been attempted in previous error slice discovery methods. As a result, there are no direct baseline methods available for comparison.

---

### Meta-Review · Area_Chair_KReL · 2024-12-11

**Metareview:**

This paper introduces DebugAgent, which a fully automated, closed-loop debug framework targeting error slice discovery problem.  DebugAgent first generates task-specific visual attributes to highlight instances prone to errors through an interpretable and structured process. It then employs an efficient slice enumeration algorithm to systematically identify error slices, overcoming the combinatorial challenges that arise during slice exploration. Extensive experiments across multiple domains, including image classification, pose estimation, and object detection, show that DebugAgent not only improves the coherence and precision of identified error slices, but also significantly enhances the model repair capabilities.

The paper is generally written in a clear way, and the topic is also valuable in model design.

**Additional Comments On Reviewer Discussion:**

All reviewers show positive scores to this paper. They also show some minor concerns, such as lack of baselines and the lack of objective evaluation metrics. However, considering the value of the motivation of this work, I recommend an acceptance. I hope the authors can carefully take reviewers' comments if the paper is finally accepted.

---

### Decision · Program_Chairs · 2025-01-22

Accept (Poster)